# Natural Co-Infections of *Aeromonas veronii* and Yellow Catfish Calicivirus (YcCV) in Ascites Disease Outbreaks in Cultured Yellow Catfish: An Emerging Fish Disease in China

**DOI:** 10.3390/ani14223289

**Published:** 2024-11-15

**Authors:** Shuai Xu, Wenli Huang, Tao Zheng, Shan Jin, Zigong Wei, Bo Guan

**Affiliations:** State Key Laboratory of Biocatalysis and Enzyme Engineering, School of Life Sciences, Hubei University, Wuhan 430062, China; xus970924@gmail.com (S.X.); 15972365509@163.com (W.H.); 15926638594@163.com (T.Z.); jinshan20032003@126.com (S.J.)

**Keywords:** yellow catfish, Yellow Catfish Calicivirus (YcCV), *Aeromonas veronii*, ascites disease, co-infection

## Abstract

The yellow catfish is an important freshwater aquaculture species in China. In recent years, this species has been prone to frequent disease outbreaks, among which ascites is a common syndrome in the aquaculture process. Ascites can be caused by a variety of pathogens, which are generally believed to be mainly bacterial. Once the disease breaks out, it is difficult to implement timely and effective control measures. Current research indicates for the first time that ascites in yellow catfish may be caused by a co-infection of bacteria and viruses, which could exacerbate the severity of diseases encountered in the aquaculture of yellow catfish.

## 1. Introduction

The yellow catfish is recognized as one of the most crucial aquaculture species in China, with a production of 565,000 tons in 2020 [1]. In China, the main cultured varieties of yellow catfish are the all-male yellow catfish (*Tachysurus fulvidraco*) and the hybrid yellow catfish (*Tachysurus fulvidraco* ♀ × *Tachysurus vachelli* ♂), with the hybrid yellow catfish being predominant [1,2,3]. The main diseases in yellow catfish aquaculture are bacterial diseases and parasitic diseases, which frequently occur from May to September in China. The primary bacterial pathogens include *Edwardsiella ictaluri* and *Aeromonas veronii* etc., while the key parasitic pathogens are *Trichodina* spp., *Ichthyophthirius multifiliis*, and *Chilodonella cyprinid* etc. [4,5,6]. *Aeromonas veronii*, known for its strong environmental adaptability and high infectivity, is a significant pathogen causing ascites disease and ulcer syndrome of yellow catfish [4]. Widely distributed in soil and water, *Aeromonas veronii* presents high infectivity and mortality rates across various regions, substantially impacting yellow catfish aquaculture in China and leading to significant economic losses. Furthermore, the extensive use of antibiotics to combat *Aeromonas veronii* infections may raise concerns about environmental and food safety in yellow catfish aquaculture.

From spring to summer in 2020, a highly contagious emerging disease outbreak firstly occurred in major yellow catfish culture areas in China, including Hubei, Sichuan, Chongqing, Hunan, Guangxi, and Guangdong, leading to substantial economic losses. This emerging disease, characterized by hemorrhages on the head, mouth, lower jaw, and fin bases and a cumulative mortality rate exceeding 90%, was resistant to water disinfection and oral antibacterial treatments [7]. The researchers identified a novel RNA virus from the Calicivirus family as the causative agent, naming it Yellow Catfish Calicivirus (YcCV) [7]. Since 2020, YcCV outbreaks have occurred annually at the transition from spring to summer, causing significant losses to the yellow catfish aquaculture. Environmental factors such as sudden changes in water temperature and the deterioration of aquaculture water are considered important causes of the YcCV outbreak in yellow catfish.

Historically, the ascites disease of yellow catfish has been predominantly attributed to bacterial infections over the past two decades. To date, there have been no reports of simultaneous bacterial and viral infections in the yellow catfish. This study is the first to document the co-infection of YcCV and *Aeromonas veronii* in ascites cases of yellow catfish.

## 2. Materials and Methods

### 2.1. Case History

Located adjacent to Futou Lake in Hubei Province, China, this aquaculture cooperative spans approximately 200 hectares, comprising soil ponds. The aquaculture cooperative specializes in farming the hybrid yellow catfish (*Tachysurus fulvidraco* ♀ × *Tachysurus vachelli* ♂). From May to July 2022, there had been a significant daily mortality of juvenile yellow catfish, with some ponds experiencing outbreaks of mass mortality. The water temperature in ponds ranged from 23 °C to 26 °C. The afflicted fish primarily displayed symptoms like ulcers, abdominal dropsy, and body surface hemorrhage, characterized by floating at the water surface, swimming near the pond edge, loss of balance, and lethargy. Since diseased fish were often accustomed to moving around the edges of the pond, conventional plunge nets were taken. Subsequently, the water was cooled and oxygenated, and the experimental fish were transported indoors. Samples were collected from 40 live diseased fish (body length 11.8 ± 1.1 cm, weight 30.6 ± 6.9 g) for pathogen analysis.

### 2.2. Bacteria Isolation and Identification

Initially, the surfaces of yellow catfish with ascites were disinfected using 75% alcohol on an ultra-clean bench, and the ascites samples were collected. These ascites samples were then inoculated onto brain heart infusion (BHI) agar plates and incubated at 30 °C for 12 h. This process was followed by 2–3 cycles of separation and purification to isolate pure strains of bacteria. The classification of the bacteria was determined through Gram staining. The purified bacteria were further cultured in BHI liquid medium at 30 °C with agitation at 150 r/min overnight. Subsequently, the bacteria were harvested through centrifugation, and their total DNA was extracted. The 16S rRNA gene was amplified with a pair of universal primers (27F: 5′-AGAGTTTGATCCTGGCTCAG-3′ and 1492R: 5′-GGTTACCTTGTTACGACTT-3′) [8], while the gyrB gene was amplified with a pair of universal primers (UP1: 5′-GAAGTCATCATGACCGTTCTGCAYGCNGGNGGNAARTTYGA-3′, and UP2r: 5′-AGCAGGGTACGGATGTGCGAGCCRTCNACRTCNGCRTCNGTCAT-3′) [9]. The annealing temperature of PCR were 58 °C for the 16S rRNA gene and 63 °C for the gyrB gene. The PCR products were analyzed by 0.8% agarose gel electrophoresis, and successful amplifications were sent for sequencing. The sequence data were compared against the NCBI database for strain classification based on homology sequence alignment. Furthermore, the 16S rRNA and gyrB gene sequence of the representative strains were selected for phylogenetic analysis using the neighbor-joining method in MEGA11, supported by 1000 bootstrap replicates, maximum composite likelihood model, uniform rates, and pairwise deletion.

### 2.3. Virus RNA Extraction and Detection

Total RNA was extracted from various organs of each affected hybrid yellow catfish, including head kidney, spleen, heart, gill, ascites, liver, brain, and intestine, using TRIzol™ Reagent (ThermoFisher Scientific, Waltham, MA, USA), as per the manufacturer’s instructions. For cDNA synthesis, reverse transcription (RT) was performed using an RT reagent Kit (Takara, San Jose, CA, USA) with the specific primer YCCV-NS-R (5′-AAAGTTGGGTGGATTGGTTTGTCAT-3′). This cDNA served as a template for nested PCR, initially using primers YCCV-NS-Fout (5′-CGCCTAAAGTTCTCCTCCTTGTTGG-3′) and YCCV-NS-R, targeting a 523 bp fragment, followed by YCCV-NS-Fout and YCCV-NS-Rin (5′-TCTGGGTAGGTGATTGGATGTGATG-3′), for a 347 bp fragment. The PCR conditions comprised 30 cycles (95 °C for 30 s, 95 °C for 15 s, 60 °C for 15 s, 72 °C for 30 s), an extension at 72 °C for 5 min, and storage at 4 °C, with products analyzed via 0.8% agarose gel electrophoresis and sequenced. The sequences were then compared against the NCBI database using BLAST (Basic Local Alignment Search Tool) to categorize the virus based on sequence homology.

### 2.4. Histopathology and Electron Microscopy

The head kidney and spleen are crucial immune organs in fish, where viruses commonly accumulate and replicate, making these organs ideal for detecting viral infections. Although the liver, intestine, and gills can also be affected by viral infections, they are generally not the primary targets for detection.

Head kidney and spleen tissue samples from yellow catfish with ascites were fixed in Bouin’s solution for 24 h, dehydrated in graded ethanol, cleared in xylene, and embedded in paraffin. Sections of 5 μm thickness were obtained using a rotary microtome and stained with hematoxylin and eosin (H&E). The sections were observed and photographed under a Leica DM4000B microscope (Leica Microsystems, Wetzlar, Germany).

For ultrastructural analysis, 1 mm^3^ tissue blocks from these organs were initially fixed in 25% glutaraldehyde, then in 1% OsO_4_ in PBS, followed by gradient ethanol dehydration. After baking at 60 °C for 48 h, ultrathin sections of 60 nm were prepared using a Leica EM UC7 ultramicrotome. These sections were then examined under a transmission electron microscope (HITACHI, HT7700, Tokyo, Japan) to study the ultrastructural characteristics of the diseased fish’s kidney and spleen tissues.

## 3. Results

### 3.1. Disease and Pathological Features

The diseased yellow catfish exhibited several primary symptoms, including floating at the surface, lethargic swimming, abdominal enlargement, skin ulcers, and bleeding in the head, oral cavity, gill cover, and lower jaw base (Figure 1A). The dissection of these fish revealed the presence of light yellow fluid in their abdominal cavities, swollen and pale livers, blood-depleted spleens and kidneys, and intestines containing clear, foodless liquid (Figure 1B). These symptoms are commonly associated with ascites disease in yellow catfish aquaculture in China.

Histopathological analysis showed that the head kidney and spleen were severely affected, exhibiting extensive necrosis, vacuolization, and tissue loss (Figure 1C,D). The head kidney displayed edema and a moderate to severe inflammatory cell infiltration (Figure 1C, black arrows). Furthermore, infected kidney cells displayed condensed and marginalized nuclei, glomerular swelling and necrosis, and vacuolar degeneration in renal tubular cells (Figure 1C, red arrows). The spleen showed varying degrees of necrosis and vacuoles of different sizes (Figure 1D, red arrows), indicating a significant impact on this organ.

### 3.2. Isolation of Bacteria and Sequence Analysis of 16S rRNA and gyrB Genes

Bacterial colonies derived from the ascites of diseased fish were isolated and cultured on BHI agar plates, revealing Gram-negative, rod-shaped bacteria (Figure 2A). Using these colonies, the about 1.5 kb target band of the bacterial 16S rRNA gene and the about 1.2 kb target band of the bacterial gyrB gene were successfully amplified (Figure 2B,C). Subsequent sequencing and sequence BLAST analysis of 16S rRNA gene demonstrated an approximately 99% similarity between this gene and *Aeromonas veronii* (MG063198.1, MF716712.1, MF716721.1). Phylogenetic analysis confirmed the classification of the isolated strain as *Aeromonas veronii*, which shared genetic homology with *Aeromonas hydrophila* (Figure 2D). In addition, the gyrB gene sequence of this strain was approximately 99% similarity to that of *Aeromonas veronii* (MW838077.1, MW838044.1, CP002607.1). Phylogenetic analysis confirmed the classification of the isolated strain as *Aeromonas veronii*, which shared genetic homology with *Aeromonas hydrophila* and *Aeromonas sobria* (Figure 2E).

### 3.3. Virus Detection and Characterization

A 347 bp segment of YcCV was successfully amplified through RT-PCR (Figure 3A). The sequencing results confirmed 100% homology with the NCBI GenBank database sequence (MZ065194), providing evidence of YcCV infection in yellow catfish. The analysis of spleen and head kidney samples from 40 yellow catfish exhibiting ascites symptoms revealed a YcCV infection rate of 77.5% (31/40). The virus was primarily detected in the kidneys and spleens, but was also present in the gill, heart, brain, liver, intestine, and ascites of some fish (Figure 3A). Furthermore, transmission electron microscopy examination of head, kidney, and spleen cells from yellow catfish illuminated the presence of abundant spherical virus particles in the cytoplasm, measuring approximately 35 nm in diameter. These observations strongly indicate an infection caused by YcCV (Figure 3B–E, red arrows).

## 4. Discussion

The increasing scale and intensification of yellow catfish farming in China have led to escalating challenges posed by disease outbreaks, particularly those caused by bacterial infections. Among these, ascites syndrome has emerged as one of the major threats to the yellow catfish aquaculture due to its widespread occurrence, high contagiousness, and significant mortality rate. Ascites disease, a complex and multifactorial condition commonly observed in fish farming, presents significant challenges in terms of effective control, primarily due to the involvement of various causative pathogens. As a result, fish farmers incur substantial economic losses. The principal pathogens associated with ascites syndrome in yellow catfish include *Aeromonas veronii*, *Aeromonas hydrophila*, and *Edwardsiella ictaluri*, among others [3,10,11]. *Aeromonas veronii*, a conditional pathogen, plays a significant role in the prevalence of ascites and ulcer syndromes of yellow catfish, producing virulence factors such as aerolysin, adhesin, cytotoxin, enterotoxin, and hemolysin [4,12,13]. Presently, the management of bacterial diseases heavily relies on the use of chemical disinfectants and antimicrobials, raising concerns about the emergence of drug-resistant strains and environmental pollution.

Prior to 2020, there were almost no reports of large-scale outbreaks of viral diseases in the yellow catfish aquaculture industry in China. However, since 2020, a new viral disease caused by the YcCV has emerged annually during the transition from spring to summer [7]. The epidemiology and pathogenesis of this virus still require comprehensive investigation. Due to the rapid evolution of caliciviruses, it is crucial to thoroughly examine their disease mechanisms and comprehend the interactions between the virus and aquatic environments, as well as other pathogens. This knowledge is essential for the development of effective control methods.

Since 2020, a large number of yellow catfish deaths have occurred annually during the spring and summer at the Futou Lake aquaculture cooperative, within an area of approximately 3000 mu, resulting in significant losses. The primary causes of these losses have been identified as diseases such as ascites, skin ulcers, and head erosion. Previous beliefs attributed these diseases to bacterial infections. However, this study has successfully confirmed that diseases such as ascites and skin ulcers in yellow catfish are caused by a combined infection of YcCV and *Aeromonas veronii*, as determined through PCR and sequencing analysis. Co-infections commonly arise when two or multiple different pathogens infect the same host (such as fish, etc.), either as simultaneous or as secondary concurrent infection [14]. Notably, previous research has demonstrated that Aeromonas veronii and YcCV individually cause necrosis in the spleen and head kidney of yellow catfish [7,15]. The findings of this study, based on symptoms, clinical signs, and histopathological results, strongly indicate a synergistic interaction between Aeromonas veronii and YcCV, amplifying their pathogenicity. Similarly, co-infections involving viruses and bacteria have been reported in tilapia farming, such as Iridovirus with bacteria (*Aeromonas veronii*, *Flavobacterium columnare*) and Tilapia Lake Virus (TiLV) with various *Aeromonas* species [16,17,18,19,20,21]. These co-infections can significantly impact host susceptibility, disease progression, severity, and duration. In various fish species, co-infections between homogeneous or heterogeneous pathogens often exhibit a synergistic effect, leading to enhanced pathogenicity [16,17,18,19,20,21,22,23]. In some cases, antagonistic interactions between homologous pathogens have been observed, resulting in the suppression of one or both pathogens [24,25]. However, synergistic effects are more commonly observed in co-infections among fish pathogens, with fewer instances of antagonistic effects [25]. Certainly, further design of biostatistical experiments is required to confirm the scientific data on the enhanced mortality rate and severity of disease in yellow catfish due to the co-infection of YcCV and *Aeromonas veronii*. Thus, further research is essential to understand the mechanisms of co-infections involving viruses and bacteria in yellow catfish.

## 5. Conclusions

Previous studies have shown that both *Aeromonas veronii* and YcCV pose significant threats to Chinese yellow catfish aquaculture. It is the first documented instance of establishing a correlation between outbreaks of ascites disease and the co-infection of *Aeromonas veronii* and YcCV in yellow catfish aquaculture. The synergistic effect resulting from the co-infection of these pathogens likely aggravates the severity of the disease. These research findings have significant implications for the accurate diagnosis of pathogens, causing ascites, skin ulcers, and other diseases in yellow catfish. Furthermore, these findings will guide the formulation of effective strategies for disease prevention and control, ultimately minimizing economic losses in the aquaculture industry.

## Figures and Tables

**Figure 1 animals-14-03289-f001:**
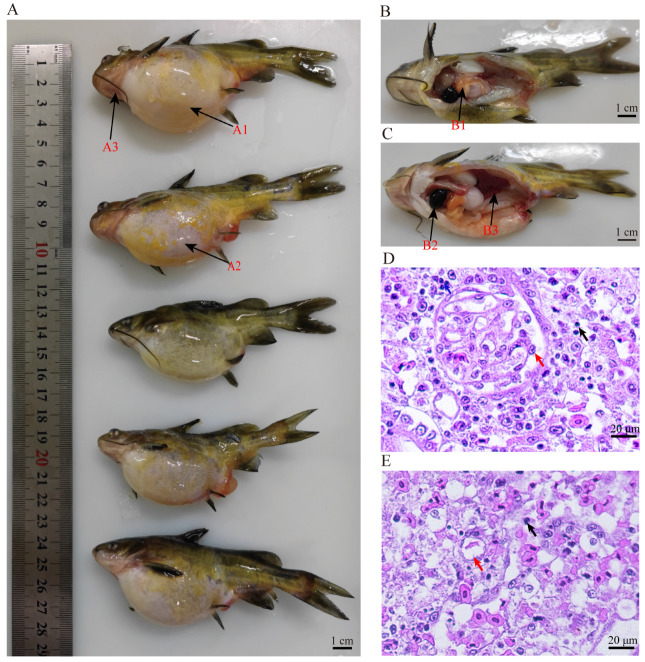
The clinical signs and histopathological analysis of infected yellow catfish. (**A**) The presence of ascites (A1), skin ulceration (A2), and bleeding in the head, oral cavity, gill cover, and lower jaw base (A3). (**B**,**C**) Noticeable swelling and paleness in the liver (B1), loss of blood in the spleen (B2) and kidneys (B3), and the presence of transparent liquid in the intestines. (**D**) The head kidney shows edema and a moderate to heavy infiltration of lymphocytes (black arrows), and kidney cells displayed condensed and marginalized nuclei, glomerular swelling, and necrosis (red arrow). (**E**) The spleen of diseased fish exhibits edema and a moderate to heavy infiltration of lymphocytes (black arrows), along with various-sized vacuolation (red arrow).

**Figure 2 animals-14-03289-f002:**
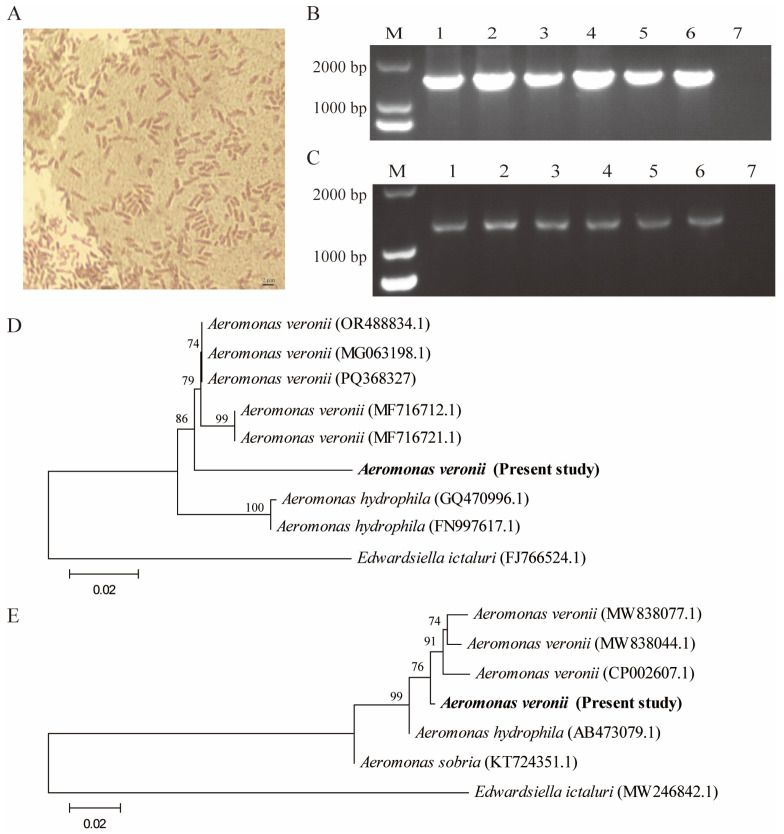
The identification of pathogenic bacteria. (**A**) Gram staining procedure. (**B**) PCR amplification of the 16S rRNA gene using universal 27F/1492R primer pair. (**C**) PCR amplification of the gyrB gene using the UP1/UP2r primer pair. (**D**) Phylogenetic tree based on the sequences of 16S rRNA genes with *Edwardsiella ictaluri* (FJ766524.1) as an outgroup. (**E**) Phylogenetic tree based on the sequences of gyrB genes with *Edwardsiella ictaluri* (MW246842.1) as an outgroup. Bootstrap values (based on 1000 replicates) > 70% are given at the branch points. Sequence accession numbers are given in parentheses.

**Figure 3 animals-14-03289-f003:**
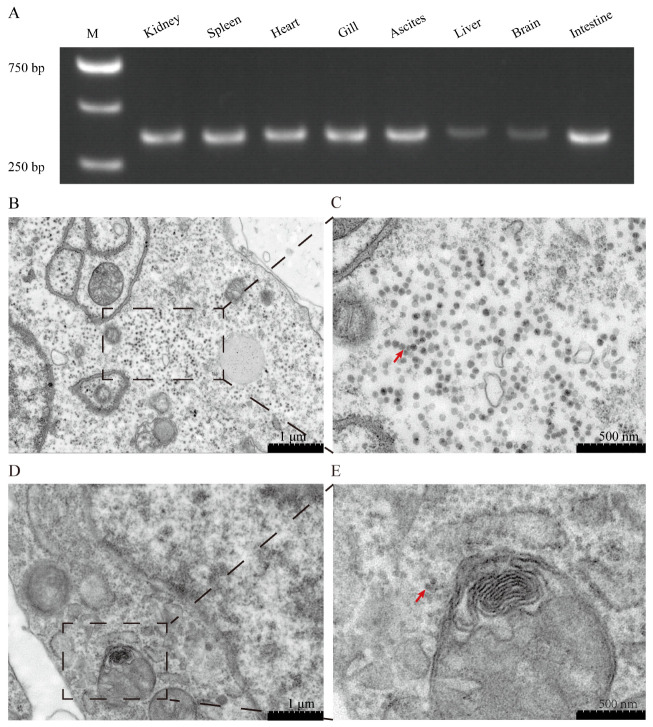
The identification of pathogenic viruses. (**A**) Distribution of YcCV in various tissues of naturally infected yellow catfish. The presence of YcCV was detected in the head kidney, spleen, heart, gill, ascites, liver, brain, and intestine tissues. (**B**,**D**) Enlarged view of the region outlined by the black rectangle, displaying magnified virus particles. (**C**) Viral particles (red arrow) observed in the cytoplasm of the head kidney. (**E**) Viral particles (red arrow) observed in the cytoplasm of the spleen.

## Data Availability

The data used to support the findings of this study are included within the article. The data will be made available on request.

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
