# Peer review of "Natural Co-Infections of Aeromonas veronii and Yellow Catfish Calicivirus (YcCV) in Ascites Disease Outbreaks in Cultured Yellow Catfish: An Emerging Fish Disease in China"

_animals, 2024, doi:10.3390/ani14223289_

Round 1

Reviewer 1 Report

Comments and Suggestions for Authors

This paper has certain innovative and practical value for the research in the field of yellow catfish disease, but there are some deficiencies in expression and argument, it is recommended to consider accepting after minor revision.

Minor concerns: 

1.     In line 75, the middle of 25-40 needs to be changed from - to , and there needs to be a space between the unit of measurement and the number

2.     In line 81, for 2-3 read 23

3.     Line 97, main text is gills, figure 3A is gill, please double check.

4.     In the text, replace hours with h and minutes with min.

5.     Line 122, the secondary heading should read Disease and Pathological Features

6.     I think figure 2 should be moved below to the section in chapter 3.2, also need to reorder it.

Author Response

Reviewer 1

Comments 1: [This paper has certain innovative and practical value for the research in the field of yellow catfish disease, but there are some deficiencies in expression and argument, it is recommended to consider accepting after minor revision.]

Response 1: [Thank you for pointing this out. We agree with this comment.]

Minor concerns:

Comments 2: [In line 75, the middle of 25-40 needs to be changed from - to –, and there needs to be a space between the unit of measurement and the number.]

Response 2: [Thank you. We have corrected this minor error in the revised manuscript.]

Comments 3: [In line 81, for 2-3 read 2–3.]

Response 3: [Thank you. We have corrected this minor error in the revised manuscript.]

Comments 4: [ Line 97, main text is gills, figure 3A is gill, please double check.]

Response 4: [Thank you. We have corrected this minor error in the revised manuscript.]

Comments 5: [ In the text, replace hours with h and minutes with min.]

Response 5: [Thank you. We have corrected this minor error in the revised manuscript.]

Comments 6: [ Line 122, the secondary heading should read Disease and Pathological Features.]

Response 6: [Thank you. We have corrected this minor error in the revised manuscript.]

Comments 7: [I think figure 2 should be moved below to the section in chapter 3.2, also need to reorder it .]

Response 7: [Thank you. We have corrected this minor error in the revised manuscript.]

Reviewer 2 Report

Comments and Suggestions for Authors

This study described ascites in yellow catfish is caused by both A. veroni and YcCV. The information presented in this article can be very useful in field. The authors well addressed the main issue in the introduction. There some minor issues the authors should address.

Title: Scientific names should be italicized. Also, use abbreviated scientific name after the first appearance.

Annual production of yellow catfish differs in the abstract and introduction.

The examined fish were below 50 g, but they are stated as “sub-adult”. Correct it.

It should be clarified if the fish transported to the laboratory alive or dead? Explain the procedure of fish capture and transportation. Also, from how many farm the samples were collected?

This study fail to answer this question: is this disease caused by A. veroni and YcCV con-infection?! For this, the authors should do an experimental challenge. If they have not this information, it should be discussed in the manuscript.

Author Response

Reviewer 2

Comments 1: [This study described ascites in yellow catfish is caused by both A. veroni and YcCV. The information presented in this article can be very useful in field. The authors well addressed the main issue in the introduction. There some minor issues the authors should address.]

Response 1: [Thank your very much for your professional comments and precious suggestion. We agree with this comment. We have corrected these minor issues in the revised manuscript.]

Comments 2: [Title: Scientific names should be italicized. Also, use abbreviated scientific name after the first appearance.]

Response 2: [Thank you. We have corrected this minor error in the revised manuscript.]

Comments 3: [Annual production of yellow catfish differs in the abstract and introduction.]

Response 3: [ Yellow catfish is one of the most important aquaculture species in China, with an annual output of 565,000 tons. We have corrected this minor error in the revised manuscript.]

Comments 4: [The examined fish were below 50 g, but they are stated as “sub-adult”. Correct it.]

Response 4: [Thank you. We have changed “sub-adult” to “juvenile” in the revised manuscript.]

Comments 5: [It should be clarified if the fish transported to the laboratory alive or dead? Explain the procedure of fish capture and transportation. Also, from how many farm the samples were collected?]

Response 5: [Thank you very much for the reviewer's suggestions on the specification description of the material fish. Since diseased fish were often accustomed to moving around the edges of the pond, conventional plunge nets were taken. Subsequently, the water was cooled and oxygenated, and the experimental fish were transported indoors. In the revised draft, we made modifications according to the reviewer's suggestions.]

Comments 6: [This study fail to answer this question: is this disease caused by A. veroni and YcCV con-infection?! For this, the authors should do an experimental challenge. If they have not this information, it should be discussed in the manuscript.]

Response 6: [Thank you for the reviewer's suggestions. This article reports the first discovery of A. veroni and YcCV co-infection in yellow catfish pond farming, presenting a new challenge for disease control in yellow catfish. The end of the article mentions content related to "Certainly, further design of biostatistical experiments is required to confirm the scien-tific data on the enhanced mortality rate and severity of disease in yellow catfish due to co-infection of YcCV and Aeromonas veronii. Thus, further research is essential to understand the mechanisms of co-infections involving viruses and bacteria in yellow catfish."]

Reviewer 3 Report

Comments and Suggestions for Authors

The manuscript by Xu et al. presents valuable documentation of the natural co-infection of Yellow Catfish Calicivirus (YcCV) and Aeromonas veronii in cultured yellow catfish (Tachysurus fulvidraco). However, the article requires substantial revisions before it can be considered for publication.

Major comments:

  1. The use of 16S rRNA gene sequencing for the identification of Aeromonas species is suboptimal. As highlighted by Janda et al. (2010), more reliable housekeeping genes, such as gyrB and rpoD, offer improved species-level resolution. It is recommended that the authors sequence the gyrB or rpoD genes of the Aeromonas isolate to confirm species identity accurately.
  2. The manuscript would benefit from the inclusion of a challenge experiment to fulfill Koch’s postulates, which are essential for confirming the causative role of both pathogens in the disease outbreak.

Minor comments:

  1. Introduction section: The authors should elaborate on whether any other viruses significantly affect yellow catfish. Discussing the major viral diseases impacting catfish, in general, would enhance the introduction.
  2. Line 75: Please provide the fish length used in this study.
  3. Lines 67–76: Include details on the physicochemical conditions of the water in which the fish were cultured, as these parameters are crucial for understanding the environmental factors involved.
  4. Lines 77–94: Clarify whether parasitological examinations were performed on the fish, as co-infections with parasites could further complicate the disease dynamics.
  5. Line 110: While histopathological analyses were performed on the kidney and spleen, the rationale for excluding other organs (e.g., liver, intestine, gills) should be provided.
  6. In Figure 2, an outgroup should be included to improve the clarity and interpretation of the phylogenetic analysis.
  7. Additionally, a phylogenetic tree based on the viral sequence (YcCV) should be drawn to support the virus's identification and relatedness to other known caliciviruses.

Author Response

Reviewer 3

Comments 1: [The manuscript by Xu et al. presents valuable documentation of the natural co-infection of Yellow Catfish Calicivirus (YcCV) and Aeromonas veronii in cultured yellow catfish (Tachysurus fulvidraco). However, the article requires substantial revisions before it can be considered for publication.]

Response 1: [Thank your very much for your professional comments and precious suggestion.]

Major comments:

Comments 2: [The use of 16S rRNA gene sequencing for the identification of Aeromonas species is suboptimal. As highlighted by Janda et al. (2010), more reliable housekeeping genes, such as gyrB and rpoD, offer improved species-level resolution. It is recommended that the authors sequence the gyrB or rpoD genes of the Aeromonas isolate to confirm species identity accurately.]

Response 1: [Thank your very much for your professional comments and precious suggestion. Following your valuable advice, we supplemented gyrB sequecing to confirm the species in the revised manuscrpt.]

Comments 2: [The manuscript would benefit from the inclusion of a challenge experiment to fulfill Koch’s postulates, which are essential for confirming the causative role of both pathogens in the disease outbreak.]

Response 2: [Thank your very much for your professional comments and precious suggestion. This article reports the first discovery of A. veroni and YcCV co-infection in yellow catfish pond farming, presenting a new challenge for disease control in yellow catfish. The end of the article mentions content related to "Certainly, further design of biostatistical experiments is required to confirm the scien-tific data on the enhanced mortality rate and severity of disease in yellow catfish due to co-infection of YcCV and Aeromonas veronii. Thus, further research is essential to understand the mechanisms of co-infections involving viruses and bacteria in yellow catfish."]

Minor comments:

Comments 3: [Introduction section: The authors should elaborate on whether any other viruses significantly affect yellow catfish. Discussing the major viral diseases impacting catfish, in general, would enhance the introduction.]

Response 3: [Thank your very much for your professional comments and precious suggestion. During the farming of yellow catfish in China, there were no previous reports of viral infections in yellow catfish. It wasn't until 2022 that the first yellow catfish virus—Yellow Catfish Calicivirus (YcCV)—was discovered.]

Comments 4: [Line 75: Please provide the fish length used in this study.]

Response 4: [Thank your very much for your professional suggestion. We recalculated the growth-related data and standardized the expression as follows: body length is 11.8±1.1 cm, and body weight is 30.6±6.9 g.]

Comments 5: [Lines 67–76: Include details on the physicochemical conditions of the water in which the fish were cultured, as these parameters are crucial for understanding the environmental factors involved.]

Response 5: [Thank your very much for your professional comments and precious suggestion..Unfortunately, at that time we did not measure the specific water quality parameters of each pond, only the water temperature parameters were tested. From May to July 2022, there had been a significant daily mortality of juvenile yellow catfish, with some ponds experiencing outbreaks of mass mortality. The water temperature in ponds ranged from 23°C to 26°C. Environmental factors such as sudden changes in water temperature and the deterioration of aquaculture water are considered important causes of the YcCV outbreak in yellow catfish. In the revised draft, we made modifications according to the reviewer's suggestions.]

Comments 6: [Lines 77–94: Clarify whether parasitological examinations were performed on the fish, as co-infections with parasites could further complicate the disease dynamics.]

Response 6: [Thank your very much for your professional comments and precious suggestion.Based on the clinical symptoms of yellow catfish, we preliminarily infer that the ascites may be caused by bacteria. Meanwhile, the poor efficacy of antibiotic treatment leads us to suspect a possible viral infection. Although parasitic infections may exacerbate fish diseases, large-scale outbreaks in yellow catfish farming are generally not attributed primarily to parasites. However, in subsequent research, we will explore this issue in reference to the reviewers' comments.]

Comments 7: [Line 110: While histopathological analyses were performed on the kidney and spleen, the rationale for excluding other organs (e.g., liver, intestine, gills) should be provided.]

Response 7: [Thank your very much for your professional comments and precious suggestion. The head kidney and spleen of fish are important immune organs. Viruses often accumulate and replicate in these immune organs, making the kidneys and spleen ideal choices for detecting viral infections. Compared to other organs, the lesions in these organs are usually more pronounced. Analyzing these two organs makes it easier to confirm the presence of viruses and assess the severity of the infection. In contrast, the liver, intestine, and gills have relatively weaker direct immune functions. Although they may also be affected by viral infections, they are generally not the preferred objects for detection.]

Comments 8: [In Figure 2, an outgroup should be included to improve the clarity and interpretation of the phylogenetic analysis.]

Response 8: [Thank your very much for your professional comments and precious suggestion. In Figures 2D and 2E, we used sequences of Aeromonas hydrophila, Aeromonas sobria, and Edwardsiella ictaluri as outgroups to enhance the clarity and interpretability of the phylogenetic analysis.]

Comments 9: [Additionally, a phylogenetic tree based on the viral sequence (YcCV) should be drawn to support the virus's identification and relatedness to other known caliciviruses.]

Response 9: [Thank your very much for your professional comments and precious suggestion.Currently, research on viral diseases in yellow catfish is relatively scarce. YcCV is the only circovirus recently discovered in yellow catfish, and Liu et al. conducted a systematic analysis of its evolutionary relationships. (Liu, W.; Xue, M.; Yang, T.; Li, Y.; Jiang, N.; Fan, Y.; Meng, Y.; Luo, X.; Zhou, Y.; Zeng, L. Characterization of a Novel RNA Virus Causing Massive Mortality in Yellow Catfish, Pelteobagrus fulvidraco, as an Emerging Genus in Caliciviridae (Pi-cornavirales). Microbiol Spectr 2022, 10, e00624-00622, doi:10.1128/spectrum.00624-22.).]

Reviewer 4 Report

Comments and Suggestions for Authors

This manuscript presents novel and timely research on the co-infection of Aeromonas veronii and YcCV as a cause of ascites disease in yellow catfish. The study is significant in challenging the previously held assumption that ascites is primarily a bacterial disease, instead highlighting the role of viral-bacterial co-infections in exacerbating disease severity. These findings are relevant to economically important yellow catfish farming regions, providing new insights into aquatic disease mechanisms.

The manuscript is generally well-structured, with a clear introduction, methodology, results, and discussion. Given its focused and escriptive, it is appropriately submitted as a short communication. However, minor revisions are required before it can be accepted for publication. Specifically, the introduction and discussion sections contain redundant statements about the economic importance of yellow catfish in Chinese aquaculture. Streamlining these sections will improve clarity and readability, ensuring that the manuscript’s context is effectively conveyed. Furthermore, the text inconsistently uses the terms "yellow catfish" and "hybrid yellow catfish" without clarification. To avoid confusion, the terminology should be consistent, or distinctions between the two strains should be explicitly defined.

Author Response

Reviewer 4

Comments 1: [This manuscript presents novel and timely research on the co-infection of Aeromonas veronii and YcCV as a cause of ascites disease in yellow catfish. The study is significant in challenging the previously held assumption that ascites is primarily a bacterial disease, instead highlighting the role of viral-bacterial co-infections in exacerbating disease severity. These findings are relevant to economically important yellow catfish farming regions, providing new insights into aquatic disease mechanisms.]

Response 1: [Thank your very much for your professional comments and precious suggestion.]

Comments 2: [The manuscript is generally well-structured, with a clear introduction, methodology, results, and discussion. Given its focused and escriptive, it is appropriately submitted as a short communication. However, minor revisions are required before it can be accepted for publication. Specifically, the introduction and discussion sections contain redundant statements about the economic importance of yellow catfish in Chinese aquaculture. Streamlining these sections will improve clarity and readability, ensuring that the manuscript’s context is effectively conveyed. Furthermore, the text inconsistently uses the terms "yellow catfish" and "hybrid yellow catfish" without clarification. To avoid confusion, the terminology should be consistent, or distinctions between the two strains should be explicitly defined.]

Response 2: [Thank your very much for your professional comments and precious suggestion. 1. In the revised manuscript, we simplified the description of the economic importance of yellow catfish in aquaculture in China within the introduction section. 2. Yellow catfish refers to all four species within the genus Tachysurus. Currently, in China, the main cultivated species of yellow catfish are the common yellow catfish (Tachysurus fulvidraco) and the hybrid yellow catfish (Tachysurus fulvidraco♀ × Tachysurus vachelli♂), with the hybrid species being predominant. The subject of this study is the hybrid yellow catfish, where co-infection with viruses and bacteria has been observed. However, common yellow catfish can also be infected with YCCV virus. To avoid confusion, we have made adjustments to the terminology, which helps improve the readability of the paper. ]

Round 2

Reviewer 3 Report

Comments and Suggestions for Authors

The authors have satisfactorily addressed all comments and the manuscript can be accepted for publication.